# BoRA: Bayesian Hierarchical Low-Rank Adaptation for Multi-Task Large Language Models

Simen Eide[1] and Arnoldo Frigessi[2]

[1]University of Oslo, Schibsted
[2]University of Oslo

## Abstract

This paper introduces Bayesian Hierarchical Low-Rank Adaptation (BoRA), a novel method for fine-tuning multi-task Large Language Models (LLMs). Current fine-tuning approaches, such as Low-Rank Adaptation (LoRA), perform exceptionally well in reducing training parameters and memory usage but face limitations when applied to multiple similar tasks. Practitioners usually have to choose between training separate models for each task or a single model for all tasks, both of which come with trade-offs in specialization and data utilization.

BoRA addresses these trade-offs by leveraging a Bayesian hierarchical model that allows tasks to share information through global hierarchical priors. This enables tasks with limited data to benefit from the overall structure derived from related tasks while allowing tasks with more data to specialize. Our experimental results show that BoRA outperforms both individual and unified model approaches, achieving lower perplexity and better generalization across tasks. This method provides a scalable and efficient solution for multi-task LLM fine-tuning, with significant practical implications for diverse applications.

## 1 Introduction

Large Language Models (LLMs) have become popular models for solving a wide range of generative text problems. LLMs are typically pretrained on large and diverse datasets of texts, and then fine-tuned on new datasets dedicated to specific tasks. The fine-tuning process is often done by maximizing the next token log likelihood of the task-specific dataset, and the resulting model is then used to make predictions and generations for new data. A popular fine-tuning method is Low-Rank Adaptation (LoRA), which reduces the number of trainable parameters by a factor of up to 10,000 and the GPU memory requirement by a factor of 3, while still outperforming other fine-tuning techniques ([1]). When performing fine-tuning of LLMs, the practitioner often wants to solve multiple similar tasks at the same time. For example: A media organization might want to use an LLM to generate various textual elements like headlines, keywords, and summaries for a given news article. A website might have multiple chatbot tasks answering user inquiries across various topics. And finally, a scriptwriting AI might want to generate dialogues for different characters in a movie, each with its unique style and vocabulary.

In these multi-task cases, the practitioner can either (1) train one model for each task and dataset separately, or (2) train one model for all tasks and condition the specific output wanted (e.g. [2]). There is a trade-off between these two approaches: Train each task separately and the model will be able to specialize on each task, but it may suffer when limited data are available for some tasks. Train one model for all tasks, and the model will be able to share information between tasks, but it may not be able to specialize on each task due to limited model capacity.

In this paper, we suggest a way to circumvent these trade-offs and present a generalization of the two LoRA fine-tuning methods for multi-task problems, which we call the Bayesian Hierarchical Low-Rank Adaptation method (BoRA). The BoRA method allows tasks to share information between each other through a set of global hierarchical prior parameters, in a Bayesian hierarchical setting. This means that tasks with little data can benefit from the global structure across related tasks, while tasks with more data can specialize on their specific tasks. We show in Section 3.4 that BoRA can be seen as a generalization of the two approaches above.

To test our method, we evaluate BoRA on the next token generation problem of parliamentary speeches, letting each parliamentary representative be a task. In our setting, BoRA is superior to both conventional approaches described above and suggests a way to mitigate the trade-off between the two. The paper is organized as follows: Section 2 presents related works. Section 3 presents BoRA, the Hierarchical LoRA method; Section 4 describes the experimental setup, Section 5 presents the results, and Section 6 discusses the results and the implications of the new method. Finally, we make the code openly available at `https://github.com/simeneide/bora`.

Proceedings of the 6th Northern Lights Deep Learning Conference (NLDL), PMLR 265, 2025.

## 2 Related Works

We group related work into two areas: multi-task learning and fine-tuning methods of large language models. Autoregressive LLMs and Bayesian modeling are described in Section 3.

### 2.1 Multi-task Learning

Multi-task Learning [3] is a popular machine learning approach for training one model on multiple tasks. A common approach is to share the lower layers of a neural network between tasks and have separate parameters for the last (or last few) layers for each task. Our paper suggests a different approach, where we share information between tasks through a hierarchical prior over the task-specific parameters.

Modern training of language models can itself be considered a form of multi-task learning where the model is trained on a diverse set of tasks and datasets [4, 5]. However, with the trend of even bigger models, pretraining of LLMs is only available to a few large organizations due to both computational and data requirements. Therefore, there is a need for methods that can leverage the pretraining of these models for smaller tasks and datasets such as the one we present in this paper.

Model merging of LLMs [6–8] is another popular approach, often showing promising performance. It takes multiple task-specific models and merges them into one model. The standard way is by doing a weighted average of the model parameters ([6]), but more sophisticated methods exist ([7, 8]). Model merging differs from our approach in that it merges the models after training, while the BoRA method shares information between tasks during training through a Bayesian hierarchical model."

### 2.2 Fine-tuning Methods

A popular approach to fine-tuning neural networks involves adjusting only the top layer (or the n top layers) of a pre-trained neural network (e.g., [9]). These techniques exploit the fact that features learned in the lower layers of the network are general and transferable, while the top layers are more task-specific. This approach has been widely adopted in neural networks. However, in this paper, we focus on the LoRA method, which instead allows us to train a low-rank decomposition of the entire network.

There are several variations of the LoRA algorithm. The Sparse Mixture of Low Rank Adapters (SIRA) [10], for instance, creates a Mixture of Expert on the low-rank parameters. Another variation, the Weight Decomposed Low-Rank Adaptation (DoRA) [11], decomposes the low-rank parameters into scale and unit norm direction parameters. There is also the Efficient Low Rank Adaptation (LoRA+) [11],

which improves the existing LoRA optimization algorithm by adjusting the learning rate of the low-rank parameters. Lastly, MoRA [12] decomposes the low-rank parameters using learnable square matrices with non-parameter based compressions and decompressions. All these methods are designed to improve the performance of language models similar to LoRA, and they can be used in conjunction with the proposed Bayesian Hierarchical LoRA method.

## 3 Method: Hierarchical LLM

This section describes the Hierarchical LoRA method for multi-task problems. We begin by defining the task and data structure, followed by an introduction to pretrained autoregressive language models. Next, we describe the LoRA method, and finally, we present the proposed Bayesian Hierarchical LoRA model.

### 3.1 Task and Data Structure

We define our study across $D$ distinct tasks, denoted by $d = 1, 2, ..., D$. For each task $d$, we have a dataset $\mathcal{D}_d$ containing $N_d$ documents, with each document $n = 1, ..., N_d$ consisting of a sequence of $W_n$ tokens. This dataset structure can be formally represented as:

$$\mathcal{D} = \{\mathcal{D}_d\}_{d=1}^D = \{\{w_{d,n,1:W_n}\}_{n=1}^{N_d}\}_{d=1}^D \quad (1)$$

In this notation, $w_{d,n,1:W_n}$ represents the token sequence in the $n^{th}$ document of the $d^{th}$ task. For simplification in subsequent discussions, we will omit the indices $d$ and $n$ (and write $w_i$) when their reference is evident from the context.

### 3.2 Pretrained Autoregressive Language Model

Our foundational model is a pretrained autoregressive language model, characterized by its parameters $\theta_{full}$ and its architecture, referred to as a Large Language Model (LLM). The parameters $\theta_{full}$ were acquired through pretraining on a large and diverse dataset, distinct from our current dataset $\mathcal{D}$. The model's probability distribution is expressed as:

$$P(w_{1:W}|\theta_{full}) = \prod_{i=1}^{W-1} LLM(w_{i+1}|w_{1:i}) \quad (2)$$

where $LLM(w_{i+1}|w_{1:i})$ denotes the probability of the next token $w_{i+1}$ given the preceding tokens $w_{1:i}$ in the document. For simplicity, we omit $\theta_{full}$ explicitly in the LLM, since the autoregressive language model inherently depends on these parameters.

## 3.3 Low-Rank Adaptation (LoRA)

A Large Language Model consists of numerous linear layers. The LoRA method reduces the number of trainable parameters by reparameterizing the weight parameters ($W \in \mathbb{R}^{n_1 \times n_2}$, $W \subset \theta$) of these layers into the original pretrained weights $W_{full}$ and a low-rank decomposition $A$ and $B$ ([11]):

$$W = W_{full} + \frac{\alpha}{r} BA \qquad (3)$$

where $W_{full} \in \mathbb{R}^{n_1 \times n_2}$ is the original pretrained weight, $B \in \mathbb{R}^{n_1 \times r}$ and $A \in \mathbb{R}^{r \times n_2}$ are the low-rank factors, $n_1$ and $n_2$ are the input and output dimensions of the original weight matrix in the linear layer, $r \ll \min(n_1, n_2)$ is the low-rank dimension, and $\alpha$ is a scaling hyperparameter.

During LoRA fine-tuning, the low-rank factors $A$ and $B$ are learned (estimated) from the data, while the original pretrained weights $W_{full}$ remain fixed to their pretrained values.

## 3.4 The Bayesian Hierarchical LoRA Model

In the hierarchical LoRA model, we introduce a LoRA parameter set for each task $d$, noted as $\theta_d$, which contains all the low-rank decomposition matrices $A_d$ and $B_d$ from each decomposed layer in the Large Language Model. The likelihood of the dataset $\mathcal{D}$ given the task parameters $\theta_{1:D}$ is then formulated by combining equations 1 and 2:

$$\mathrm{L}(\mathcal{D}|\theta_1, \theta_2, \ldots, \theta_D) = \prod_{d=1}^{D} L(\mathcal{D}_d|\theta_d)$$

$$= \prod_{d=1}^{D} \prod_{n=1}^{N_d} \prod_{i=1}^{W_n - 1} \mathrm{LLM}(w_{d,n,i+1}|w_{d,n,1:i}; \theta_d). \qquad (4)$$

Tasks share some common knowledge, modeled by introducing a Gaussian hierarchical prior over the task parameters $\theta_{1:D}$, described by:

$$P(\theta_{1:D}|\Theta, \tau) = \prod_{d=1}^{D} N(\theta_d; \Theta, \frac{1}{\tau} I) \qquad (5)$$

where $\Theta$ is a set of hierarchical mean parameters, $I$ is a unit diagonal matrix, and $\tau \geq 0$ is a scalar precision hyperparameter controlling how similar the task parameters $\theta_d$ should be to the hierarchical mean parameters $\Theta$. We let $\Theta$ have an improper uniform hyperprior (i.e. $P(\Theta) = 1$ for all values of $\Theta$). The prior assumes that the task parameters share a common origin because the tasks are related: input data come from similar distributions and the output tasks share similarities. The precision hyperparameter $\tau$ dictates how much the task parameters $\theta_d$ are a priori expected to deviate from the hierarchical mean parameters $\Theta$. It indicates how much

structure and information is shared among tasks and should be set by the practitioner based on prior knowledge. The posterior distribution of the hierarchical model, obtained by combining equations 4 and 5, is:

$$P(\theta_{1:D}|\mathcal{D}, \tau) \propto \prod_{d=1}^{D} \mathrm{L}(\mathcal{D}_d|\theta_d) \cdot P(\theta_d|\Theta, \tau) \qquad (6)$$

The resulting BoRA model utilizes each task based on its dataset size and similarity to other tasks: If a task has few data points, the posterior in equation 6 will be dominated by the prior term, and the task parameters will remain near the global hierarchical parameters, borrowing information from these. Conversely, if a task has many data points, the likelihood will dominate the posterior, allowing the task parameters to focus more on the individual task.

The hierarchical LoRA model can be seen as a generalization of the two conventional approaches for multi-task problems: If we let $\tau$ approach zero, the probability distribution of the prior (eq. 5) will become a flat uniform prior and the situation will be equivalent to training each task independently. On the other hand, if we let $\tau$ approach infinity, the prior will become a strong constraint on the task parameters, preventing them from deviating from the global parameters, and the situation will be equivalent to training one model for all tasks.

## 3.5 Optimization

Given the precision hyperparameter $\tau$, we seek the maximum a posteriori (MAP) estimate of the hierarchical model. We achieve this using AdamW ([13]), a gradient-based optimization algorithm, by optimizing over the task parameters $\theta_{1:D}$ and the hierarchical mean parameters $\Theta$ using the posterior in equation 6. This implies that the task parameters will have gradients both towards minimizing the conventional LLM loss (represented by the likelihood term) and the hierarchical mean parameters (originating from the prior). The hierarchical mean parameters $\Theta$ will have gradients towards the average of the task parameters, learning to represent the global structure of the tasks.

An important observation is that the gradient of the prior term in equation 6 is proportional to the precision hyperparameter $\tau$. This means that the optimization will require more steps to converge for larger values of $\tau$, as the task parameters will be more constrained towards the hierarchical mean parameters and a posteriori dependent on each other. Therefore, we adjust the learning rate of the AdamW optimizer to be proportional to the precision hyperparameter $\tau$, and we find in our experiments that this causes the optimizations to converge at a similar rate for different values of $\tau$.

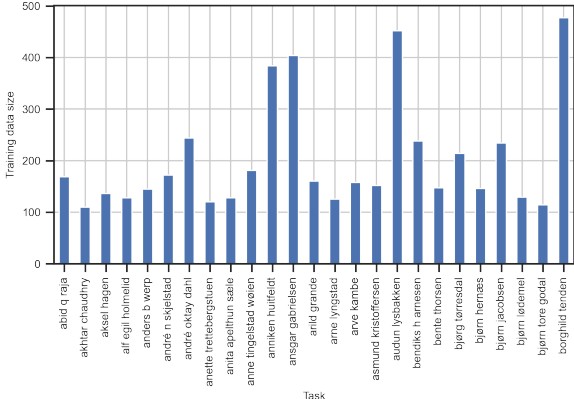

**Figure 1.** Number of training examples for each task.

where all task-specific model parameters are constrained to be equal ($\tau = 10000$).

| Learning Rate | Precision ($\tau$) | Test Perplexity |
|:---:|:---:|:---:|
| $10^{-4}$ | 0 | 16.80 |
| $10^{-5}$ | $10^0$ | 16.59 |
| $10^{-4}$ | $10^1$ | 12.85 |
| $10^{-3}$ | $10^2$ | **12.82** |
| $10^{-2}$ | $10^3$ | 13.26 |
| $10^{-1}$ | $10^4$ | 13.91 |

**Table 1.** Empirical results of the hierarchical LoRA fine-tuned on the three tasks. The perplexity is calculated on the test set.

# 4 Experimental setup

To test our method, we optimize the model with a range of different precision hyperparameters $\tau$, and evaluate the perplexity on the test set for each task. Perplexity is defined as the exponentiated average negative log-likelihood per token, and is a common metric for evaluating language models. A lower perplexity indicates a higher likelihood and a better model. We use the Talk of Norway Dataset [14], which consists of speeches by Norwegian parliament politicians. Speakers come from different political parties and geographical areas, but the tasks will still share a common domain represented by parliamentary speeches in Norway over a short period of time. Therefore, we can consider different speakers of this dataset as different tasks. Specifically, we select the first 25 speakers chronologically who have more than 100 speeches in the dataset. This results in 25 tasks with samples ranging from 110 to 477 documents, and an average of 202 documents per task. For some speakers, we have significantly less data compared to others. The distribution of samples per task can be seen in Figure 1. We reserve 33% of the data for the test set.

We select a model that has been pretrained on a large and diverse dataset, but has not seen the specific tasks that we want to evaluate it on. We use the 'opt-350m' model [15], which is predominantly trained on English, and has only seen a small amount of Norwegian data from the CommonCrawls dataset.

# 5 Results

The overall results of the hierarchical LoRA model on the Talk of Norway Dataset are shown in Table 1 and Figure 2. We see that the model achieves the best test perplexity of 12.82 when using a precision parameter of 100. The improvement is largest compared to training the models independently ($\tau = 0$), and smaller when comparing to the limiting case

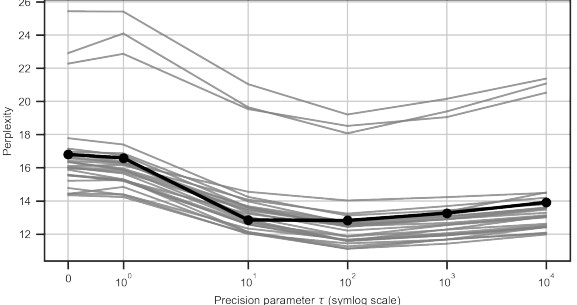

**Figure 2.** Plot of Precision vs Perplexity. The thick black line is the overall test perplexity across all tasks, and the thinner grey lines represent the test perplexity for each individual task. The leftmost point corresponds to training each task independently ($\tau = 0$), and the rightmost point corresponds to the limiting case when all task-specific model parameters are constrained to be equal ($\tau \to \infty$).

When considering each task separately (Table A.1), we see that all tasks benefit from a hierarchical model compared to both the case when all models are trained independently ($\tau = 0$) and the limiting case where all task-specific model parameters are constrained to be equal ($\tau = 10000$). Figure 3 shows the relative improvement of each task when using the hierarchical model compared to the two cases versus their dataset sizes. There is an indication that the improvement from training separately is larger for tasks with less data, while this is not observed for the "one model" case. A precision parameter of 100 gives the best perplexity for all our tasks.

# 6 Discussion

Our results in Figure 2 demonstrate that the hierarchical LoRA model can improve test perplexity for all tasks compared to both training the models independently and training one model for all tasks. This confirms our hypothesis that BoRA can be a useful method for multi-task problems, mitigating

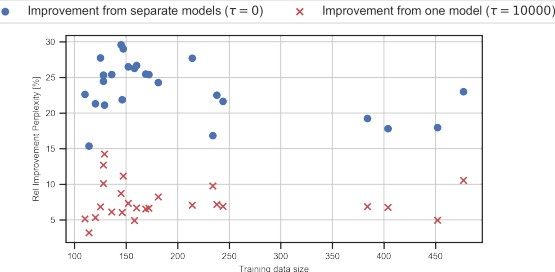

**Figure 3.** Task dataset size versus the relative improvement in perplexity of the best performing hierarchical model ($\tau = 100$) compared to the case when all models are trained independently ($\tau = 0$) in blue, and compared to the limiting case when all task-specific model parameters are constrained to be equal ($\tau = 10000$) in red. The figure shows that all tasks benefit from sharing parameters with the global model, and indicates that tasks with less data benefit more than those with more data.

the trade-off between training each task separately and training one model for all tasks. Notably, BoRA also benefits tasks with relatively more data. We observe in Figure 4 that the distance between the task parameters and the global prior increases with increasing task dataset size. This is expected in a hierarchical model, as tasks with more data will have a larger likelihood term in the posterior, allowing them to specialize more on their specific task during training.

We observe no consistent relationship between dataset size and the final perplexity (Figure 5). Although one might expect that tasks with more data would learn more about their specific task and achieve lower perplexity, this pattern is not evident in our experiments. Other task-specific factors, such as under-represented dialects and political views, might influence the final perplexity. Controlling for these factors is complex and beyond the scope of this paper, but they are likely contributors to this outcome. However, when considering the relative improvement of our best-performing model, the results in Figure 3 indicate that tasks with less data may benefit more than those with more data. Although the signal is subtle, this is an expected result, as the hierarchical model allows tasks with limited data to leverage information and structure from the global model, while tasks with more data can specialize in their specific task.

In principle, one could adopt a fully Bayesian approach, instead of Maximum a Posteriori (MAP) maximization, to assess the reliability of the hierarchical model. Such an approach would necessitate the use of Markov Chain Monte Carlo (MCMC), Variational Inference, or other methods.

This approach would involve sampling from the posterior distribution of the hierarchical model, al-

lowing for the estimation of uncertainty in the task parameters. However, MCMC methods are computationally intensive, and it remains unclear to us how to efficiently and scalably sample from the posterior distribution of the language model. A middle ground between a full Bayesian approach and the MAP estimate could involve being Bayesian for select parameters. For example, instead of our current approach where we maximize the likelihood of the validation set concerning the precision parameter $\tau$, one could be Bayesian and integrate over its empirical posterior distribution obtained from MCMC. This would provide insights into how well tasks share information in a specific problem context. We leave these explorations for future research.

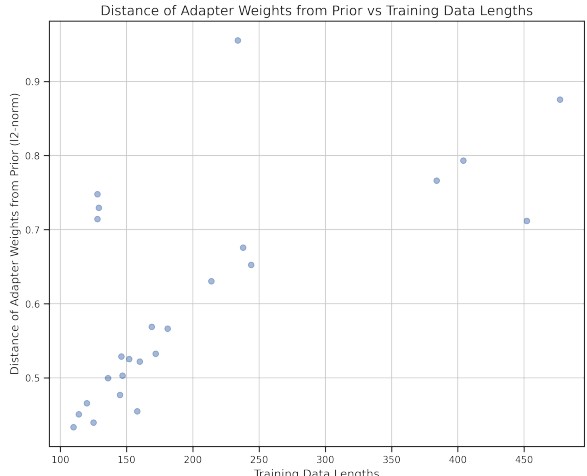

**Figure 4.** The figure shows the L2-distance between each task's adapter weights and the global prior on the y-axis, and the number of training examples on the x-axis. As expected, tasks with more training data have a larger distance to the global prior.

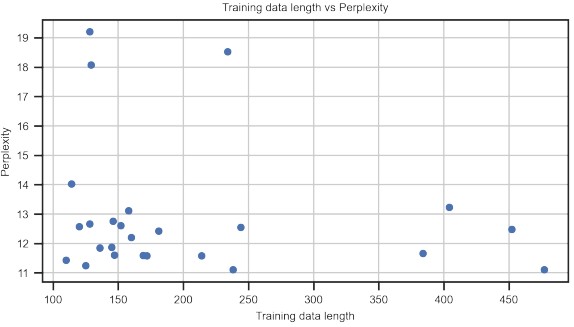

**Figure 5.** Training size vs. perplexity of the best-performing hierarchical model ($\tau = 100$). The figure shows that there is no strong relationship between dataset size and final perplexity.

# 7 Conclusion

This paper presents a novel generalization of the LoRA fine-tuning method for multi-task problems, which we call BoRA, Bayesian Hierarchical Low-Rank Adaptation for Multi-task Large Language Models. The hierarchical model allows tasks to share information between each other through a set of global hierarchical prior parameters and can be seen as a generalization of the two conventional approaches for multi-task problems. We demonstrate that BoRA can improve test perplexity for all tasks compared to both training the models independently and training a single model for all tasks, effectively mitigating the trade-off between these two approaches. We propose adjusting the learning rate according to the design parameter that controls the trade-off between likelihood and prior. Future work includes investigating the effect of BoRA on more tasks and datasets, examining how the relative capacity of the global model affects performance, and exploring the full posterior distribution of the model.

## 7.1 Funding

This work was supported by Schibsted Media group and the Research Council of Norway, Integreat - Norwegian Centre for knowledge-driven machine learning, project number 332645.

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

# A Task-Specific Test Perplexities

Table A.1 shows the test perplexities for each task when using BoRA with different precision hyperparameters $\tau$.

**Table A.1.** Test perplexity for each task and precision hyperparameter $\tau$

| tau
Task | 0 | 1 | 10 | 100 | 1000 | 10000 |
|---|---|---|---|---|---|---|
| abid q raja | 15.56 | 15.20 | 12.57 | **11.60** | 11.91 | 12.41 |
| akhtar chaudhry | 14.78 | 14.37 | 12.01 | **11.43** | 11.66 | 12.06 |
| aksel hagen | 15.88 | 15.30 | 12.68 | **11.84** | 12.26 | 12.61 |
| alf egil holmelid | 16.96 | 16.86 | 13.97 | **12.66** | 13.35 | 14.51 |
| anders b werp | 16.86 | 16.35 | 13.04 | **11.87** | 12.54 | 13.00 |
| andré n skjelstad | 15.53 | 15.23 | 12.59 | **11.58** | 11.98 | 12.41 |
| andré oktay dahl | 16.01 | 15.85 | 13.50 | **12.55** | 12.88 | 13.48 |
| anette trettebergstuen | 15.98 | 15.73 | 13.43 | **12.57** | 12.90 | 13.28 |
| anita apelthun sæle | 25.44 | 25.42 | 21.04 | **19.21** | 20.16 | 21.37 |
| anne tingelstad wøien | 16.41 | 16.23 | 13.60 | **12.43** | 12.90 | 13.54 |
| anniken huitfeldt | 14.44 | 14.39 | 12.33 | **11.66** | 12.05 | 12.53 |
| ansgar gabrielsen | 16.10 | 16.16 | 14.07 | **13.23** | 13.68 | 14.19 |
| arild grande | 16.65 | 16.29 | 13.31 | **12.21** | 12.56 | 13.08 |
| arne lyngstad | 15.57 | 15.28 | 12.12 | **11.25** | 11.66 | 12.08 |
| arve kambe | 17.78 | 17.40 | 14.24 | **13.11** | 13.43 | 13.79 |
| asmund kristoffersen | 17.15 | 16.75 | 13.63 | **12.61** | 13.03 | 13.61 |
| audun lysbakken | 15.21 | 15.25 | 13.22 | **12.48** | 12.67 | 13.13 |
| bendiks h arnesen | 14.34 | 14.23 | 12.04 | **11.11** | 11.42 | 11.97 |
| bente thorsen | 16.35 | 15.88 | 12.87 | **11.60** | 12.28 | 13.06 |
| bjørg tørresdal | 16.02 | 15.65 | 12.78 | **11.58** | 11.92 | 12.46 |
| bjørn hernæs | 16.33 | 15.94 | 13.68 | **12.76** | 13.08 | 13.58 |
| bjørn jacobsen | 22.28 | 22.87 | 19.54 | **18.52** | 19.05 | 20.53 |
| bjørn lødemel | 22.91 | 24.09 | 19.65 | **18.08** | 19.39 | 21.08 |
| bjørn tore godal | 16.57 | 16.30 | 14.55 | **14.02** | 14.22 | 14.48 |
| borghild tenden | 14.42 | 14.84 | 12.12 | **11.10** | 11.68 | 12.41 |

