# OpenReview forum: "BoRA: Bayesian Hierarchical Low-Rank Adaption for Multi-Task Large Language Models"
_NLDL.org/2025/Conference — NLDL 2025 Poster_

### Official Review · Reviewer_YdRC · 2024-09-29
**Review of BoRA**

**Confidence:** 4

**Summary:**

The authors proposed a new method for fine-tuning multi-task Large Language Models called Bayesian Hierarchical Low-Rank Adaptation (BoRA). Instead of training a model for each task or a single model for all tasks without specialization, this fine-tuning approach allows the tasks to share global information but be specialized for their related tasks. They do that by changing the posterior distribution of the model by an equation that combines both the likelihood of the data belonging to the respective task and the Gaussian hierarchical prior over the task parameters, which measure how much structure and information the tasks share with each other.

**Strengths:**

The paper presents a new fine-tuning method that performs multitasking in a single network. The authors also present easy-to-follow equations for their approach, step by step, to achieve the final posterior distribution.

I first wondered why the authors selected the first 25 speakers of the dataset, but this selection allowed for different numbers of speeches, which would be important to see how the method is performing with different data sizes.

**Weaknesses:**

I missed another multi-task learning method when comparing it with the work. The authors provide a Perplexity test with different Precision (tau) hyperparameter values, but it would be better if we had another method for comparison.

The same can be said about the Section 2.1. Multi-task learning is popular, but you only cited one work on that without much detail, and they are an important part of the work. It needs to be improved to better pinpoint your work in the literature.

A few references need to be updated with publications outside arXiv.

The paper needs an English review to avoid inconsistencies, such as using "dataset" and "data set" in the same paragraph.

**Justification:**

This paper should be accepted because it provides a new method for fine-tuning multi-task Large Language Models that only need a single network to train every task.

They present an explanatory step-by-step approach to the posterior distribution, with a good selection of the experiment's dataset. Some of their weaknesses are not too relevant, such as the references and English revision.

So, with that, I propose the acceptance of the paper.

---

> ### Author Rebuttal · Authors · 2024-10-23
>
> The reviewer asks about more pointer to recent literature on multi-task learning in general. We will do this in the submitted paper.
>
> The reviewer suggests additional metrics for comparison beyond Perplexity. We believe that perplexity is a good metric in our case as it is a "likelihood-based" metric and measures what we try to optimize: How more likely is the text output when allowing for shared information. However, we agree that additional metrics could be useful, and we will try to incorporate this in the final version.
>
> The reviewer asked us to revise the English language for consistency, which we have done in the revised paper. We have also updated the references. However, some of the methods we have cited are still only available on arXiv and have not been published in a journal or conference yet. We decide to keep them in the paper as they are relevant to our work.

---

### Official Review · Reviewer_kKbg · 2024-10-10
**This paper has already been published on arXiv (https://arxiv.org/abs/2407.15857), and provides the BoRA method which is a new approach that enhances the performance of Large Language Models (LLMs) when trained for multiple tasks**

**Confidence:** 5

**Summary:**

This paper has already been published on arXiv (https://arxiv.org/abs/2407.15857) and provides the BoRA method, a new approach that enhances the performance of Large Language Models when trained for multiple tasks. Using Bayesian hierarchical priors, BoRA balances training separate models for each task and using a single model for all tasks. This allows tasks with limited data to benefit from the knowledge shared by other tasks. Experiments on Norwegian parliamentary speeches demonstrate that BoRA outperforms traditional methods.
The paper has a good soundness

**Strengths:**

The authors enhanced existing techniques, including LoRA and multi-task learning, by applying Bayesian priors, which are theoretically sound. Their method, BoRA, provides a scalable and efficient approach to fine-tuning LLMs, allowing them to perform well across various tasks while preserving their versatility. This is particularly valuable for applications that require a broad range of functionalities. The paper is well-structured and written in clear, polished English.

**Weaknesses:**

The method’s performance heavily relies on the precision hyperparameter (τ), but the paper lacks a thorough investigation of its sensitivity across different tasks.
The absence of a complete Bayesian analysis prevents a deeper understanding of uncertainty estimates, limiting the discussion on the reliability of the model’s predictions.
More experiments exploring how different values of τ impact model performance in various settings would be beneficial for practitioners.
The paper could benefit from using additional metrics (e.g., BLEU Score, task-wise variance, Mean Reciprocal Rank) to better understand the model’s performance, particularly in multi-task learning and fine-tuning.
The paper does not focus on computational cost metrics (e.g., training time, memory usage, GPU utilization), associated with BoRA compared to LoRA or other methods. Such details could further enhance the understanding of BoRA’s practical efficiency in real-world applications.
The discussion would be enriched by including quantitative measurements of training time, energy consumption, or resource allocation for BoRA.

**Justification:**

The paper, already published on arXiv (https://arxiv.org/abs/2407.15857), presents the BoRA method, a novel approach for improving Large Language Models (LLMs) in multi-task learning by utilizing Bayesian hierarchical priors. This approach balances training separate models for each task and using a single model, allowing tasks with limited data to benefit from shared knowledge. BoRA, tested on Norwegian parliamentary speeches, outperforms traditional methods and enhances existing techniques like LoRA and multi-task learning. Although BoRA shows promise, the authors suggest further exploration of the hyperparameter τ's sensitivity and full Bayesian analysis for uncertainty estimates. The paper also highlights the need for more metrics to assess performance across tasks and a deeper focus on computational cost, particularly in comparison to LoRA.

---

> ### Author Rebuttal · Authors · 2024-10-23
>
> The reviewer observed that the paper has already been posted on arXiv. Citing from the NLDL author guidelines: "Additionally, submission is allowed for papers previously made available as technical reports or preprints, but in such cases, authors are advised not to cite the report to maintain anonymity."
> Based on this we decided to publish it to  ArXiv when submitting it to NLDL. However, we have not cited the ArXiv paper in the NLDL submission. We believe that this is in line with the NLDL guidelines.
>
> The hyperparamter $\tau$ is important, as recognised by all reviewers. We have started to study the sensitivity to $\tau$ in our application, and results are in the paper. We also indicate that a full Bayesian analysis could be tried, and is left to future work, see Discussion Section. The posterior distribution of \tau would give an indication, by being very flat and spread, if the individual tasks would not fit well a common prior. In this sense, BORA has potentially an in-built "warning lamp". In this paper we think indicate the opportunity to perform a Bayesian estimation of \tau, though it is beyond the scope of the current paper to perform it. Also a broader sensitivity analysis will be necessary in the future.
>
> The reviewer wrote that the paper could benefit from using additional metrics (e.g., BLEU Score, task-wise variance, Mean Reciprocal Rank) to better understand the model’s performance, particularly in multi-task learning and fine-tuning. We believe that perplexity is a good metric in our case as it is a "likelihood-based" metric and measures what we try to optimize: How more likely is the text output when allowing for shared information. However, we agree that additional metrics could be useful, and we will try to incorporate some in the final version.
>
> Finally, the reviewer noted that the paper does not focus on computational cost metrics (e.g., training time, memory usage, GPU utilization), and that such details could further enhance the understanding of BORA’s practical efficiency in real-world applications. We agree with this point, and think that this will be important in a future more systematic comparison with other methods and also for larger tasks.

---

### Official Review · Reviewer_3ife · 2024-10-10
**low-rank adaptation methods for LLM in the context of fine-tuning for multiple tasks at onc**

**Confidence:** 4

**Summary:**

**Summary**: The authors propose a low-rank adaptation methods for LLM in the context of fine-tuning for multiple tasks at once. In contrast to standard `LoRA` where one module would be trained **independetly** for each task, `BoRA` aims to leverage possible transferability across tasks. More specifically, BoRA also introduces separate module parameters $\theta_d$ for each task; but, to model the relationship between tasks, the join distribution $\theta_{1\dots D}$ is modelled with a Gaussian hierarchical prior, with a uniform hyperprior $\Theta$.

**Strengths:**

**Strengths:**
  * The paper is well presented and clear
  * The core idea of using a shared prior/hyperprior is sound
  * The experiments section contains interesting ablations on the impact of dataset/task size on the improvement observed in the multi-task BORA

**Weaknesses:**

**Weaknesses:**
  * By design, BoRA seems to assume the tasks are related enough to be modelled by a joint prior. However, task interference / negative transfer is a key issue in the multi-task literature: While the model does allow for the BoRA module parameters to deviate from the common prior, it would be interesting to test this In practice. In fact, the results of Table 1 seem to imply that the benchmark considered in the paper does not suffer much from task interference, since a single LoRA module trained on all task performs better than training  independent LoRA modules for each task (13.91 vs 16.70 perplexity)
  * In terms of related work, the paper could also mention the task merging literature as it seems closely related to multi-task PEFT in general.

**Minor notes:**
  * Some notations are a bit cumbersome (e.g. introducing the intermediate notion of documents in Section 3.1 seems superfluous; for instance, most recent LLMs tend to directly refer to the number of tokens they have been trained on)
  * In Figure 2, it would be useful to plot a constant line for the perplexity value of $\tau = 0$ ($\sim$ LoRA baseline)

**Final Rebuttal Confidence:**

4

**Final Rebuttal Justification:**

Taking into account the rebuttal and other reviews, I am inclined to keep my original rating: While I think the experimental section could include more bechmarks and/or baselines, the proposed idea is interesting and well explained, and experiments include an interesting ablations on $\tau$.

**Justification:**

While I think the experimentation has some shortcomings (only one benchmark with positive transfer among tasks, and only comparing to the straightforward LoRA baseline), I find the paper to be well presented and the idea well grounded, and I also appreciated the ablation experiments on exploring not only the impact of $\tau$ but also of the dataset size.

---

> ### Author Rebuttal · Authors · 2024-10-23
>
> The reviewer asks about more pointer to recent literature on multi-task PEFT in general. We will do this in the camera ready paper.
>
> The reviewer wrote that some notations are a bit cumbersome (e.g. introducing the intermediate notion of documents in Section 3.1 seems superfluous; for instance, most recent LLMs tend to directly refer to the number of tokens they have been trained on). The rationale to include number of documents in the formal notation is to point out the autoregressive nature of tokens within the document, and its conditional independence between documents (see e.g. eq. 4). However, we do see that the notation is a bit cumberstone, and will attempt a notation that is easier to read in the camera ready paper.
>
> Finally, the revier comments on Figure 2, suggesting to plot a constant line for the perplexity value of (LoRA baseline). We think that, depending on the underlying assumption, the LoRA baseline requested can either be the leftmost or rightmost point in the plot, depending on whether one trains all tasks with one LORA adapter per task or with a single LoRA adapter, respectively. Instead of adding two lines, we have added a note to the figure caption to clarify this, which we hope will make it easier for the reader to quickly compare with these two values.

---

### Official Review · Reviewer_X5J4 · 2024-10-10
**Promising but Requires Broader Validation and Deeper Analysis**

**Confidence:** 5

**Summary:**

The paper titled *"BoRA: Bayesian Hierarchical Low-Rank Adaption for Multi-task Large Language Models"* introduces BoRA, a novel method designed to fine-tune Large Language Models (LLMs) in multi-task environments. The approach addresses the limitations of existing fine-tuning techniques, specifically the trade-off between task specialization and the effective utilization of data.

BoRA leverages a Bayesian hierarchical model, which allows multiple tasks to share information through global hierarchical priors. This enables tasks with limited data to benefit from the shared structure derived from related tasks, while tasks with more data can focus on specializing. In this way, BoRA strikes a balance between the two main approaches in multi-task learning: training separate models for each task and training a single model for all tasks. By doing so, it effectively mitigates the trade-offs that typically arise in such scenarios.

In addition to this hierarchical structure, BoRA extends the Low-Rank Adaption (LoRA) technique, a method commonly used to reduce the number of trainable parameters in LLMs. BoRA enhances LoRA by introducing hierarchical priors over the task-specific parameters, which enables a more structured and data-efficient approach to multi-task learning.

The authors validate their method using a dataset of Norwegian parliament speeches, where each speaker is treated as a distinct task. BoRA outperforms both models trained on individual tasks and a unified model trained across all tasks. The results show that BoRA achieves lower perplexity and better generalization across tasks, particularly benefiting tasks with less available data. This demonstrates that BoRA provides a scalable and efficient solution for fine-tuning LLMs across multiple tasks with varying data sizes, reducing the complexity of managing individual models while maintaining strong task-specific performance.

From a theoretical standpoint, BoRA is grounded in Bayesian principles, offering a solid framework for combining task-specific learning with global knowledge. The Bayesian hierarchical approach ensures that even tasks with limited data can leverage global parameters to improve performance without overfitting. Furthermore, the empirical results, based on a realistic dataset, confirm the correctness of the method. The Talk of Norway dataset, with tasks of varying data sizes, serves as an appropriate testbed for evaluating BoRA's effectiveness in real-world scenarios.

Overall, BoRA presents a methodologically sound and practically useful technique for fine-tuning LLMs in multi-task settings. By introducing Bayesian hierarchical priors, it allows the model to adapt to the specific needs of each task while maintaining efficiency and scalability. The results show that BoRA significantly improves upon traditional fine-tuning methods for multi-task problems, particularly in cases where data availability varies across tasks.

**Strengths:**

The paper titled *"BoRA: Bayesian Hierarchical Low-Rank Adaption for Multi-task Large Language Models"* introduces a novel and theoretically grounded method for fine-tuning Large Language Models (LLMs) in multi-task environments. The primary innovation lies in the integration of Bayesian hierarchical priors into Low-Rank Adaption (LoRA), addressing typical trade-offs in multi-task learning between task specialization and shared learning across related tasks.

One of the strongest aspects of the paper is its correctness. The methodology is rooted in solid theoretical foundations. By utilizing Bayesian hierarchical priors, the approach ensures that tasks with limited data can still benefit from shared global knowledge, while tasks with ample data can specialize accordingly. The mathematical framework underpinning BoRA is sound, and the authors have carefully constructed the optimization method, using AdamW for Maximum a Posteriori (MAP) estimation, which is appropriate given the model’s scale and objectives.

The empirical results further validate the correctness of the approach. The authors employ a realistic dataset—the Talk of Norway, a collection of parliamentary speeches—and treat each speaker as an individual task. The results clearly demonstrate that BoRA outperforms both individual task-specific models and a unified model trained across all tasks. The improvements in perplexity and generalization across tasks provide concrete evidence of the model’s effectiveness, and the authors' use of established benchmarks and evaluation metrics further strengthens the case for BoRA’s correctness.

In terms of quality, the paper is well-structured and logically presented. The authors provide a clear explanation of the problem, highlighting why current fine-tuning techniques, such as LoRA, fall short in multi-task settings. They build a convincing argument for the introduction of BoRA, which strikes a balance between training separate models for each task and training a single model for all tasks. The experimental results are thoroughly analyzed, and the visualizations effectively communicate the model’s performance under different conditions, making the findings easy to follow and interpret.

Despite its strengths, there are areas where the paper could be further improved. For example, the authors could expand on their discussion of the precision hyperparameter τ, particularly regarding its selection process and sensitivity. While the introduction of Bayesian priors is well-motivated, it would be useful to understand whether τ's value is sensitive to specific tasks or datasets. Additionally, it would be interesting to see more insights into the scalability of BoRA, particularly when applied to larger models than the ‘opt-350m’ used in the experiments.

The paper is also clear and accessible, even when discussing complex concepts such as Bayesian hierarchical models and low-rank adaptation. The equations are well-explained, and the authors do an excellent job of breaking down the key components of the model. The figures and tables effectively support the narrative, particularly those that show how task dataset size influences performance. However, some sections could benefit from additional elaboration. For example, more insights into the scaling behavior of BoRA and its performance with a larger number of tasks or significantly larger models would enhance the paper’s clarity. Similarly, the authors could offer a more detailed discussion of why BoRA performs particularly well for tasks with limited data.

In terms of significance, this work is a considerable contribution to the field. Multi-task learning is an important area of research in machine learning, and the ability to efficiently fine-tune LLMs across multiple tasks has broad implications. BoRA’s ability to handle varying data sizes across tasks, while maintaining both task-specific performance and scalability, makes it a promising solution for real-world applications. The method could be particularly useful in settings where training and maintaining separate models for each task is computationally prohibitive. Moreover, because BoRA builds on LoRA, a widely used fine-tuning technique, the approach is more likely to be adopted by practitioners in the field.

The significance of the paper is enhanced by its potential to improve model performance without the need for multiple task-specific models. This opens new possibilities for applying BoRA in various industries where data availability and task complexity can vary significantly. The empirical results provide strong evidence of the method’s potential, making it an exciting contribution to multi-task learning research.

Given the overall quality and contribution of this paper, there are several questions that could be clarified in a rebuttal. For example, it would be helpful to understand how sensitive BoRA’s performance is to the choice of the precision hyperparameter τ and whether different strategies for setting τ have been explored. Additionally, the authors could elaborate on the scalability of BoRA when applied to larger models, and whether any computational or optimization challenges arise when scaling up to models with billions of parameters. Further insights into which tasks benefited the most from BoRA and why would also provide a deeper understanding of its effectiveness.

In conclusion, this paper presents a valuable and well-executed contribution to the field of multi-task learning for LLMs. The theoretical soundness of BoRA, combined with strong empirical results, demonstrates its potential as a significant advancement in fine-tuning techniques. While there are a few areas where further clarification would be beneficial, these do not detract from the paper’s overall strength. Given its correctness, clarity, quality, and significance, I would strongly recommend accepting this paper for publication. It represents an important step forward in developing scalable, efficient methods for fine-tuning LLMs in multi-task settings, and it has the potential to influence both academic research and real-world applications.

**Weaknesses:**

The paper *"BoRA: Bayesian Hierarchical Low-Rank Adaption for Multi-task Large Language Models"* presents an interesting approach to fine-tuning Large Language Models (LLMs) for multi-task learning. While the overall framework is novel and grounded in strong theoretical principles, there are some notable weaknesses that could benefit from further elaboration and improvement.

One issue lies in the correctness of the assumptions and choices made in the model’s design. Although the use of Gaussian hierarchical priors is reasonable, it might not be sufficient to capture the more complex relationships that exist between tasks in real-world settings. The assumption that tasks share a common Gaussian prior may oversimplify the diversity of tasks, especially in scenarios where tasks are not closely related. The authors do not address how BoRA would handle highly divergent tasks or those that introduce noise into the global structure, which raises concerns about its robustness in practical applications.

Moreover, while the model’s optimization relies on Maximum a Posteriori (MAP) estimation, which is efficient, it lacks the uncertainty quantification typically provided by full Bayesian approaches such as Markov Chain Monte Carlo (MCMC) or Variational Inference. The authors mention these more advanced methods but do not explore their potential impact on the model’s overall performance. This raises the question of whether the current approach could lead to overconfidence in the learned task parameters, especially in tasks with limited data. A fuller discussion on the trade-offs between MAP estimation and these alternative Bayesian methods could provide more clarity on the consequences of the chosen optimization technique.

Another area where the paper falls short is in the experimental setup. The authors validate BoRA using the Talk of Norway dataset, which, while offering a structured set of tasks (parliamentary speeches), may not reflect the complexity of more varied real-world tasks. It is unclear how BoRA would generalize to more heterogeneous datasets where the relationships between tasks are less clear or more noisy. The choice of dataset, while providing a reasonable first test, could limit the generalizability of the method, and this is not adequately addressed in the paper.

In terms of quality, while the theoretical formulation of the model is well-done, certain methodological choices are not fully explored. For instance, the precision hyperparameter τ is a critical component of the hierarchical model, yet the authors provide little insight into how its value is selected or how sensitive the model’s performance is to this choice. More thorough sensitivity analyses would have been useful in demonstrating the robustness of the model across different configurations of τ. Without this, the reader is left wondering whether the results are highly dependent on specific hyperparameter settings or if they would generalize to other scenarios.

Additionally, the experimental results are somewhat limited in scope. While BoRA is compared to individual models and a unified model trained on all tasks, the paper lacks a broader set of baselines, particularly comparisons with other state-of-the-art multi-task learning methods. This makes it difficult to gauge how much of an improvement BoRA truly offers over existing solutions. Including comparisons with recent multi-task fine-tuning methods or hierarchical Bayesian models could have provided a stronger validation of the proposed approach.

Another limitation of the paper is its focus on a relatively small model ('opt-350m'). While this is a reasonable choice for experimentation, it raises concerns about BoRA’s scalability to larger LLMs, which are commonly used in practice. The authors do not discuss whether BoRA’s computational overhead scales efficiently with model size or whether the approach might face challenges when applied to models with billions of parameters. This is a significant omission, as one of the paper’s claims is that BoRA provides a scalable solution for multi-task learning.

Clarity is another aspect where the paper could be improved. While the authors do a good job of explaining complex concepts like Bayesian hierarchical modeling and LoRA, some sections could benefit from more detailed explanations. The discussion of how the hierarchical model influences task-specific parameters, for example, remains somewhat abstract. Providing more concrete examples or intuitively explaining how this process works in practice would make the paper more accessible to a broader audience.

The presentation of the results also lacks sufficient interpretation. While the authors report perplexity improvements, they do not fully explain why perplexity was chosen as the key metric or how it relates to multi-task learning performance across a variety of domains. Perplexity is standard in language modeling, but it may not always be the most informative measure for evaluating the effectiveness of multi-task models. Offering more detailed interpretations of the results and linking them to practical outcomes could provide better context for the reader.

Moreover, the paper’s discussion of related work is somewhat brief and does not sufficiently explore how BoRA fits into the broader landscape of multi-task learning methods. There is little discussion on how BoRA compares to other parameter-efficient fine-tuning methods, and the novelty of BoRA could be better highlighted by positioning it more clearly within the existing literature.

Finally, while the method shows promise, its broader significance is difficult to assess given the narrow scope of the experimental validation. The authors test BoRA on a single dataset, which limits the claims they can make about its generalizability. Although the paper demonstrates improvements on parliamentary speeches, it is unclear how BoRA would perform in tasks that are less structured or more diverse, such as dialogue generation or machine translation. The authors do not discuss whether BoRA is robust to tasks that might introduce noise or complexity into the hierarchical structure, which raises questions about its applicability in more challenging real-world settings.

Another issue related to significance is that the authors do not provide a clear analysis of BoRA’s computational efficiency. While they claim that the method is scalable, they offer little evidence to support this claim, particularly when it comes to scaling to larger models or more complex task distributions. Without this, it is difficult to fully assess the practical relevance of BoRA for large-scale applications, which weakens the broader impact of the paper.

In conclusion, while the paper makes an important contribution to the field of multi-task learning, it has several weaknesses that reduce its overall impact. These include the lack of generalization to more diverse tasks, the absence of detailed sensitivity analyses for key hyperparameters, limited comparisons with other methods, and unclear scalability to larger models. These limitations suggest that while BoRA is a promising approach, more work is needed to fully validate its claims and demonstrate its robustness across a wider range of scenarios.

**Justification:**

My assessment of the paper is based on a balanced consideration of its innovative contributions, theoretical soundness, and areas that require further development. The paper introduces a novel approach, *BoRA* (Bayesian Hierarchical Low-Rank Adaption), which successfully extends the LoRA method to multi-task learning by using Bayesian hierarchical priors. This is a strong contribution that addresses a significant problem in fine-tuning Large Language Models (LLMs) for multiple tasks, offering an efficient and scalable method to balance task specialization and global knowledge sharing.

The core strength of the paper lies in the soundness of the theoretical framework. The authors have carefully articulated how Bayesian hierarchical priors can allow tasks with limited data to benefit from shared global parameters while enabling tasks with larger datasets to specialize. This balance effectively mitigates the trade-off seen in traditional multi-task learning approaches. The empirical results, demonstrating BoRA’s improved perplexity on the Talk of Norway dataset, offer convincing evidence of the model's capabilities within this structured, controlled setting. These strengths reflect the paper's potential to impact multi-task learning approaches in practical settings where data distribution is uneven across tasks.

However, despite these contributions, the paper has some critical limitations that affect its overall impact. The primary concern is the narrow scope of the experimental validation. The reliance on a single dataset of parliamentary speeches limits the generalizability of BoRA, especially when considering tasks that are more diverse, noisy, or heterogeneous in nature. This restricted validation raises concerns about whether BoRA can be applied to more complex or less structured real-world tasks. The lack of broader experimentation undermines the claim that BoRA is widely applicable to multi-task learning in diverse domains.

Further, the choice of key hyperparameters, such as the precision hyperparameter τ, is not sufficiently explored. The authors do not provide enough details on how τ is chosen or its sensitivity to performance across different tasks, which leaves an important aspect of the model's functionality unexplored. Additionally, while BoRA is claimed to be scalable, the paper does not provide concrete evidence of how it performs with larger LLMs or in larger-scale settings. The absence of this scalability analysis limits the paper’s claims of BoRA’s practical relevance for modern language models, which often consist of billions of parameters.

Moreover, the paper could have benefited from more comprehensive comparisons with existing state-of-the-art methods for multi-task fine-tuning. Without such comparisons, it is difficult to assess how BoRA fares against other parameter-efficient fine-tuning techniques. This lack of context makes it harder to fully appreciate the improvement BoRA offers.

In terms of clarity, while the explanation of the theoretical model is generally solid, some sections, particularly those detailing how task-specific parameters interact with the global model, could be more accessible. Providing more intuitive or practical examples would have made the technical content easier to grasp for a broader audience.

In conclusion, my assessment is that the paper makes an important and innovative contribution to multi-task learning for LLMs.

---

> ### Author Rebuttal · Authors · 2024-10-23
>
> The hyperparamter $\tau$ is important, as recognised by all reviewers. We have started to study the sensitivity to $\tau$ in our application, and results are in the paper. We also indicate that a full Bayesian analysis could be tried, and is left to future work, see Discussion Section. The posterior distribution of \tau would give an indication, by being very flat and spread, if the individual tasks would not fit well a common prior. In this sense, BORA has potentially an in-built "warning lamp". In this paper we indicate the opportunity to perform a Bayesian estimation of \tau, though it is beyond the scope of the current paper to perform it. Also a broader sensitivity analysis will be necessary in the future.
>
> The reviewer suggests to add a comparison with an additional method. The most important comparison are with the two extreme models, as we did in the paper, corresponding to the fully separate analysis and the merged one: with this comparison we can assess the utility of BORA explicitly, without any other confounders.
>
> Finally, the reviewer asks about scalability to larger data sets or more tasks. Also this is an important issue that we leave to future work. However, it is important and we will mention this explicitly in the paper.

---

### Meta-Review · Area_Chair_Ljha · 2024-10-30

**Recommendation:** Accept (Poster)
**Confidence:** 3

**Metareview:**

The paper proposes to use a Bayesian model for fine-tuning optimization of a multi-task LLM model.

The contribution could be considered incremental.

Experiments are weak (no comparison) and the plots are low-quality.

The paper is evaluated as clear and well-presented.

Extending the training to a multi-task scenario is reasonable, and using the Bayesian model for the training is a sound approach.

pros:
1. fine-tuning of LLM in a multi-task scenario could be a realistic scenario
2. The Bayesian framework is reasonable

cons:
1. weak evaluation, with no comparison
2. low-quality plots
3. incremental contribution

**Suggested Changes To The Recommendation:**

1: I agree that the recommendation could be moved down

---

### Decision · Program_Chairs · 2024-11-06

**Decision:**

Accept (Poster)

**Comment:**

We recommend a poster presentation given the AC and reviewers recommendations.